# Proteomic Profile of *M. longissimus thoracis* from Commercial Lambs Reared in Different Forage Systems

**DOI:** 10.3390/foods11101419

**Published:** 2022-05-13

**Authors:** Yangfan Ye, Evelyne Maes, Santanu Deb-Choudhury, Charles A. Hefer, Nicola M. Schreurs, Carolina E. Realini

**Affiliations:** 1AgResearch Limited, Te Ohu Rangahau Kai, Massey University Campus, Grasslands, Tennent Drive, Palmerston North 4474, New Zealand; ye.yangfan5591@hotmail.com (Y.Y.); carolina.realini@agresearch.co.nz (C.E.R.); 2Animal Science, School of Agriculture and Environment, Massey University, Private Bag 11222, Palmerston North 4442, New Zealand; n.m.schreurs@massey.ac.nz; 3AgResearch Limited, Lincoln Research Centre, 1365 Springs Road, Lincoln 7674, New Zealand; evelyne.maes@agresearch.co.nz (E.M.); charles.hefer@agresearch.co.nz (C.A.H.); 4Riddet Institute, Based at Massey University, Palmerston North 4474, New Zealand

**Keywords:** lamb, proteomics, production system, meat quality, pasture

## Abstract

This study compared the protein composition of *M. longissimus thoracis* of lambs from six commercial forage production systems in New Zealand. A total of 286 proteins were identified based on liquid chromatography-tandem mass spectrometry. First, a binomial model showed that different production groups could be distinguished based on abundances of 16 proteins. Second, pair-wise comparisons were performed to search for protein abundance differences in meat due to animal sex (ewe vs. wether), diet (perennial ryegrass vs. chicory), and age (4 vs. 6–8 months old). Greater abundance of some myofibrillar and sarcoplasmic proteins were observed in lamb loins from ewes compared to wethers. Chicory diet and older age at slaughter were associated with meat with lower abundance of some myofibrillar proteins, possibly due to a greater proportion of muscle glycolytic fibres. The proteins that showed significant differences in their abundances due to production factors could be further investigated to understand their influence on meat quality.

## 1. Introduction

Red meat exports from New Zealand generated $8.39 billion revenue in total for the 2019–2020 season [1]. Improvement in meat quality to satisfy consumers and entice repurchase is considered key for maintaining markets [2]. The meat quality attributes of lamb are influenced by several production system factors including animal age at slaughter, forage type and breed [3,4]. Increasing animal age at slaughter was reported to be favourable for meat juiciness and flavour due to greater intramuscular fat concentration, but unfavourable for tenderness [5]. Young animals produce meat that is lighter and less red in colour compared to older animals [6]. Furthermore, at the same age of slaughter, females could provide a more flavourful, tender, intense coloured meat than males [7].

Being able to modify meat quality via production systems is dependent on understanding the intrinsic components of meat that can influence key quality attributes such as tenderness, flavour, colour, and juiciness [8]. Previous studies have shown links between protein profiles and meat quality characteristics [7]. The texture and tenderness of meat are directly related to its protein matrix, and is affected by post-translational changes such as glycosylation, protein backbone cleavage, aggregation, oxidation, and protein crosslinks [9,10]. Several myofibrillar proteins, such as actin and myosin regulatory light chain 2, have been proposed as biomarkers for tenderness [11]. An important contributor towards meat colour is the abundance of myoglobin [12]. Myosin regulatory light chain 2, aldose reductase, and β-enolase were also positively associated with meat redness [13]. Troponin and myosin light chain with their dense hydrophobic regions were, however, negatively associated with water holding capacity of meat [14]. Creatine kinase has a greater expression in the muscle of animals that have experienced stress [15]. The greater expression of creatine kinase can hinder the muscle pH decline during post-mortem aging, and consequently influence the water holding capacity of meat [15]. Sheep breeds also have a role to play in the quality of meat. Merinos have a propensity to produce meat with a higher pH because they lose greater amounts of muscle glycogen than other breeds under pre-slaughter stress [3]. Therefore, the abundance of creatine kinase may also differ between different lamb breeds. Currently, however, there is still a knowledge gap on how production systems influence muscle proteins and their potential impact on meat quality. Thus, studies are required to look at the proteomic profile of meat produced under different production regimes. The broad range of forage-based commercial production systems utilized in New Zealand suggests that there is potential for variation in protein profiles of lamb meat, as a deeper understanding of these variations in protein profiles offers potential to improve meat quality. The objective of this study was to compare protein profiles of meat from six types of typical commercial New Zealand forage lamb production systems.

## 2. Materials and Methods

### 2.1. Animals and Management

To encompass a range of forage-based production systems, lambs were sourced from three commercial farms north of Invercargill, New Zealand. The six production systems (Table 1) included: 4-month-old wether lambs of a composite breed at weaning (WEAN-W), 6- to 8-month-old wether lambs of a composite breed that had been grazing perennial ryegrass based pasture (GRASS-W); 6- to 8-month-old wether and ewe lambs of a composite breed that had been grazing chicory (CHIC-W and CHIC-E, respectively); 6- to 8-month-old wether lambs of a composite breed that had been grazing red clover (REDC-W); and 12-month-old wether Merino lambs that had been grazing a mixed pasture (MXME-W). Eight lambs from each production system were randomly selected and identified at slaughter from a larger group of lambs.

### 2.2. Meat Sampling and Storage

Lambs were obtained from three commercial farms where the sex, diet, age, and genetics were known. The key aspect of this study was that the lambs were obtained from commercially relevant systems used in New Zealand. Lambs were slaughtered within 24 h after leaving the farm and underwent standard commercial processing at Alliance Group Ltd., Lorneville plant near Invercargill. The *Longissimus thoracis* (loin) muscle was removed from the carcass, vacuum packed and chilled at −1.5 °C for 21 days for aging, then frozen at −20 °C until further analysis. A detailed description of animal groups, sample collection and carcass and meat quality characteristics corresponding to this study were previously reported by Ye et al. [4]. Prior to proteomics analysis, all samples were freeze-dried at −50 °C for 2 days (Cuddon Freeze Drier, Blenheim, New Zealand) and then ground into a fine (<1 mm) powder which was stored at −80 °C until protein extraction.

### 2.3. Protein Extraction

An amount of 200 mg of frozen ground lamb tissue was soaked in 2 mL lysis buffer (7 M Urea, 2 M Thiourea, 1% freshly-added dithiothreitol and 1 protease inhibitor tablet in 50 mL buffer) and homogenised in a hand-held, serrated pestle homogeniser over ice for four cycles of 1 min grinding and 10 s pause as previously described by Yu et al. [16]. After vortexing the homogenate for 30 min at 4 °C, the insoluble material was pelleted by centrifugation of the homogenate at 15,000× *g* for 30 min at 4 °C. The pellet was removed, and the supernatant was collected and stored at −80 °C. The protein concentrations were determined using the 2D-Quant kit according to the manufacturer’s instructions (GE Healthcare, Chicago, IL, USA). A total of 150 μg protein from each samples was extracted from supernatant using chloroform-methanol precipitation method based on Wessel and Flügge [17].

### 2.4. Protein Digestion

Protein pellets were resuspended in 100 μL 0.1 M ammonium bicarbonate containing 1% sodium deoxycholate (SDC), then reduced with 20 μL of 50 mM dithiothreitol in 0.1 M ammonium bicarbonate at 56 °C for 45 min. Alkylation was performed by adding 20 μL of 150 mM iodoacetamide in 0.1 M ammonium bicarbonate solution and vortex for 30 min in the dark at room temperature (21 °C). Protein digestion was performed by adding 2 μg MS-grade trypsin (Promega, Madison, WI, USA) and 10% acetonitrile, vortexing briefly and incubating at 37 °C for 18 h. The SDC was precipitated and removed by adding 5% formic acid (FA) and centrifuged. The resulting peptides were desalted with C18 spin columns according to the manufacturer’s instructions (Thermo Scientific, Waltham, MA, USA), dried by a centrifuge concentrator and resuspended in 0.1% formic acid prior to LC-MS/MS analysis.

### 2.5. Mass Spectrometric Analysis

Liquid chromatography-mass spectrometry (LC-MS) was performed on a nanoflow Ultimate 3000 Dionex UPLC (Thermo Scientific) coupled to an Impact HD mass spectrometer equipped with a CaptiveSpray source (Bruker Daltonik, Bremen, Germany). For each sample, 1 µL of the sample was loaded on a C18 PepMap100 nano-Trap column (300 µm ID × 5 mm, 5 micron 100 Å) at a flow rate of 3000 nL/min. The trap column was then switched in line with the analytical column ProntoSIL C18AQ (100 µm ID × 150 mm 3 micron 200 Å). The reverse phase elution gradient was from 2% to 20% to 45% B over 60 min, total 85 min at a flow rate of 1000 nL/min. Solvent A was LCMS-grade water with 0.1% FA; solvent B was LCMS-grade ACN with 0.1% FA.

The LC was directly interfaced with a captive spray ion source (3.0 L/min dry gas, operated at 1500 V) to a high-resolution Impact HD quadrupole-time-of-flight (Q-TOF) (Bruker Daltonics, Bremen, Germany) mass spectrometer. To profile protein expression patterns, the analytes were detected via MS-only mode in positive ion mode, with a mass range between 130–2200 *m*/*z* and a sampling rate of 2 Hz. To link the expression levels with identifications, a pool of per treatment was created, and these pooled samples were run via LC-MS/MS with data-dependent auto-MS/MS mode with the following settings: the same LC parameters as described before, a full scan spectrum, with a mass range of 350–2200 *m*/*z*, was followed by a maximum of ten collision-induced dissociation (CID) tandem mass spectra at a sampling rate of 2 Hz for MS scans and 1 to 20 Hz for MS/MS. Precursors with charges 2+ to 3+ were preferred for further fragmentation and a dynamic exclusion of 60 s was set. Following the LC-MS run, the Q-TOF data were further analysed with Compass DataAnalysis 4.4 software (Bruker Daltonik) to evaluate the LC chromatogram and the overall quality of both MS1 and MS2 spectra.

### 2.6. Protein Identification

The PEAKS X + Studio data analysis software package (Bio informatic Solutions Inc., Waterloo, ON, Canada) was used to analyse the LC-MS/MS data. The raw data were refined by a built-in algorithm which allows association of chimeric spectra. The proteins/peptides were identified with the following parameters: a precursor mass error tolerance of 10 ppm and fragment mass error tolerance of 0.05 Da were allowed, the UNC01_Ovis aries database (v2020.06, 53326 sequences) was used, the cRAP database was used as contaminant database, semi-trypsin was specified as digestive enzyme and up to 2 missed cleavages were allowed. Carbamidomethylation of cysteine was set as fixed modification. Both oxidation (M) and deamidation (NQ) are chosen as variable modifications in Peaks DB, and in the optimized Peaks PTM search Pyro-Glu from Q, amidation, and carbamidomethylation (DHE and N-term) were added to the variable modification list. A maximum of 3 post-translational modifications (PTMs) per peptide was permitted. False discovery rate (FDR) estimation was made based on decoy-fusion. An FDR of <1% with a peptide hit threshold of −10 log *p* > 19.3 and a PTM A-score of 100 was considered adequate for confident peptide identification. To allow for confident protein identification and relative quantification, at least two unique peptides per protein were required.

### 2.7. Label Free Quantification

To quantify the protein expression levels, label-free quantification (LFQ) was performed using the quantitation node of Peaks Studio X + software. Here, expression levels between all samples were compared. The following parameters were included: a mass tolerance error of 15 ppm and a retention time shift tolerance of 2 min was allowed. To determine the relative protein and peptide abundance in the retention time aligned samples, peptide feature based quantification was performed. Relative comparison between samples is based on the area under the curve and to get this cumulative area for each protein, only unique peptides that are assigned to a particular protein were selected.

### 2.8. Statistical Analysis

The relative abundance estimates resulting from the LFQ analyses were used to select proteins with a measured abundance in at least 75% of the samples, prior to scaling to zero means and unit variances. Supervised Partial Least Square Discriminant Analysis (PLS-DA) were performed on the scaled LFQ results to identify clusters of samples using mixOmics software (version 6.12.2) [18] and R (version 4.0.2).

Eight samples from each of the six treatment groups (REDC-W, MXME-W, GRASS-W, CHIC-W, CHIC-E, and WEAN-W) were sent for mass spectrometry analysis. The samples were randomly assigned to six batches for analysis, with duplicate samples present in each batch, along with standards and quality control samples. During quality control of the samples, a distinct batch effect was detected for the samples assigned to the first batch (group B1, Appendix A). The batch effect was detected by performing an analysis of variance on the first three components of the PCA. Significant differences between the batches were detected in all three components (*p* < 0.05). From the 11 samples present in the first batch of analysis, three samples were duplicated in different batches. By removing the samples from batch 1 from the analysis, no significant batch effect was then detectable between the batches (Appendix A). After quality control, 7 samples processed in the first batch of the analysis were removed from the dataset, reducing the number of samples to 41 across the six production groups (*n* = 8 for REDC-W, *n* = 7 for MXME-W, *n* = 6 for GRASS-W, *n* = 6 for CHIC-W, *n* = 7 for CHIC-E, and *n* = 7 for WEAN-W).

A negative binomial model, with the abundance level as the response level and the interaction between the protein and each production system, was applied to identify proteins of lamb loin muscle that were significantly different between commercial forage production systems. Using the binomial model, additional pairwise comparisons were made between lamb groups differing in sex (CHIC-E vs. CHIC-W), diet (GRASS-W vs. CHIC-W), and age (CHIC-W vs. WEAN-W) for each of the proteins. For all models, *p*-values were adjusted for multiple testing using the Benjamini–Hochberg correction and *p*-values smaller than 0.05 were deemed significant.

## 3. Results

### 3.1. Differences in Protein Profiles in Meat from Lambs Reared in Six Forage Production Systems

In total, 286 proteins were identified across 41 samples with at least two unique peptides per protein. Of these, 281 proteins were detected and quantified in at least 75% of all samples. The PLS-DA plot indicated that there are some differences and also similarities between the overall protein profiles in meat from the six groups of lambs (Figure 1). The first component (x-variate) and the second component (y-variate) of the PLS-DA explained 26% and 13% of the data variation, respectively. The CHIC-W lambs were grouped separately from CHIC-E lambs in the plot, although some overlaps were observed. Among meat from all six production systems, REDC-W lambs showed the lowest variation in protein profiles measured.

A further look to see what drives these subtle differences between the protein profiles in the different groups was performed by fitting a negative binomial model to the interaction between the production systems and each protein detected. The mean abundance level for each protein was then predicted using the generated model, followed by a comparison between the group means of each production system.

Overall, 16 proteins were identified by the negative binomial model as being significantly different between the production systems (Table 2). Within the seven myofibrillar proteins, the CHIC-W lambs exhibited the most unique protein profile, with four proteins differing from all five production systems and three proteins differing from some production systems including troponin C from CHIC-E and WEAN-W, myosin-8 from REDC-W, and myosin light chain 1 from CHIC-E. Meat from CHIC-W lambs differed from CHIC-E and WEAN-W in Troponin C abundance, from REDC-W in Myosin 8 abundance and from CHIC-E in Myosin light chain 1 abundance.

Within the six sarcoplasmic proteins, a unique protein profile was observed in loins from REDC-W lambs, which differed from all other animal groups in heat shock proteins family A (Hsp70) and cognate 71 kDa abundances. The abundance of immunoglobulin lambda-1 light chain-like protein varied greatly among all production systems. Creatine kinase U-type mitochondrial protein in meat was significantly different between MXME-W lambs and the other animal groups.

The stromal proteins showed a difference in collagen proteins between lamb loins from CHIC-W and CHIC-E production systems (Table 2). A significant variation was also observed between lamb loins from different production systems for heparan sulfate proteoglycan 2.

### 3.2. Protein Comparison between the Meat Proteome from Ewe and Wether Lambs Grazed on Chicory (Chic-E vs. Chic-W)

The myofibrillar, sarcoplasmic, and stromal proteins that were significantly different between ewes and wethers that grazed on chicory are presented in Table 3. The abundances of all myofibrillar and sarcoplasmic proteins were higher in meat from CHIC-E than CHIC-W lambs, except for collagen alpha-3 chain which was more abundant in meat from wether than ewe lambs.

### 3.3. Protein Comparison between the Meat Proteome from Wether Lambs Grazed on Perennial Ryegrass and Chicory (Grass-W vs. Chic-W)

The abundances of only four myofibrillar proteins differed (Table 4) between the loins of 6- to 8-month-old wether lambs that grazed on perennial ryegrass (*Lolium perenne*) compared to chicory (*Cichorium intybus*). The abundance of alpha skeletal muscle isoform X4 of actin was 17.22-fold greater in loins from lambs grazed on perennial ryegrass than chicory. The abundance of skeletal muscle isoform of myosin regulatory light chain 2, skeletal muscle isoform X1 of myosin regulatory light chain 2, and myosin-2 were 5.22, 6.21 and 3.70-fold greater in loins from lambs grazed on perennial ryegrass than chicory, respectively.

### 3.4. Protein Comparison between Meat from Wether Lambs Slaughtered at 4-Months of Age and 6- to 8-Months of Age (Wean-W vs. Chic-W)

Alpha skeletal muscle isoform X4 of actin was 16.66-fold more abundant in the loin meat from 4-month-old pre-weaning lambs than 6- to 8-month-old lambs grazed on chicory (Table 5). The abundances of the other myofibrillar proteins in meat, Myosin regulatory light chain 2, Myosin-2, Myosin-8, and Troponin C, were also greater in loins from 4-month-old than 6- to 8-month-old lambs. Other proteins such as heparan sulfate proteoglycan was more abundant in loins from 4-month-old than 6- to 8-month-old lambs.

## 4. Discussion

The objective of the present study was to explore the protein profiles of meat from six types of typical commercial New Zealand lamb production systems that included animals of different sex, genetics, age at slaughter, and forage diets. Based on the 286 proteins that were identified across all samples, 270 proteins (Appendix A) that contributed to various functions in lamb loin muscle were similar between all six production systems. This indicated that the expressions of more than 90% of proteins identified in lamb loins in the current study are relatively stable under different forage production systems. This is in agreement with Picard et al. [7] who indicated that the abundance of proteins in cattle muscles is insensitive to diet composition, animal sex, and rearing practices. However, 16 proteins found in this study enabled different groupings of loins from lambs reared in the different production systems. These 16 proteins included structural myofibrillar proteins, sarcoplasmic proteins involved in biological pathways in the muscle related to energy metabolism and cell turnover, and stromal proteins associated with muscle cell structure.

Myofibrillar proteins are the most abundant proteins in lamb muscle, constituting 65–75% of the total muscle proteins. Myosin and actin are the major myofibrillar proteins accounting for approximate 45% and 25% of the total myofibrillar proteins in muscle [19]. Seven out of 16 of the discriminating proteins from the six production system groups belong to this class. Based on these proteins, except for troponin C, myosin-8, and myosin light chain 1, meat from CHIC-W lambs was clearly distinguished from the other five production systems.

Considering sarcoplasmic proteins, clear differences between groups were found for heat shock proteins and creatine kinase. Heat shock protein family A (Hsp70) and heat shock cognate 71 kDa protein showed a greater expression in REDC-W lambs than those from other production systems. It is not possible to elucidate the reason meat from REDC-W lambs had a greater expression of heat shock proteins from this study, which would require further investigation. Heat shock proteins are associated with impeding apoptosis and they have been reported to prevent the activity of proteolytic enzymes associated with post-mortem aging of meat [20,21,22]. As such, the expression of heat shock proteins has been correlated to a reduction in myofibrillar protein degradation and leading to meat toughness [11,21]. However, the impact of heat shock proteins expression on proteolysis did not translate into changes in meat tenderness. Previously reported shear force data from this study showed that meat from REDC-W lambs had similar (*p* > 0.05) shear force values to meat from the other types of lamb except for MXME-W, which showed higher shear force values [7]. This indicated that the differences in the abundances of these proteins may not be large enough to have a significant impact on meat quality such as tenderness for lambs slaughtered at 6 to 8 months old. Mitochondrial creatine kinase U-type in meat from MXME-W lambs differed from the other animal production groups. The main function of creatine kinase in living animals is to catalyse the reversible phosphotransferase reaction between creatine and adenosine triphosphate (ATP), which is required when high ATP regeneration is demanded. In the post-mortem muscle, creatine kinase is used to maintain ATP concentration when it cannot be regenerated through oxidative metabolism [23]. Meat from MXME-W lambs showed significantly higher muscle pH after 21 days of ageing than meat from the other animal groups in this study (5.78 vs. 5.39–5.47 mean values, respectively) [7]. This may be partly due to the different abundance of creatine kinase which has the ability to slow the decline in post-mortem muscle pH [15]. All other proteins couldn’t specifically differentiate just one production system.

To gain further insights, pairwise comparisons were performed to evaluate the effect of gender (ewe vs. wether), diet (perennial ryegrass vs. chicory), and age at slaughter (4- vs. 6- to 8-month-old) on the relative abundance of proteins in meat from animals within a single farm.

### 4.1. Proteome Abundance Differences Due to Gender (Chic-E vs. Chic-W)

Among the 4 myosin types (myosin-2, -6, -7, and -8) observed in the current study, only myosin-2 showed greater abundance in meat from ewe lambs (CHIC-E) than wether lambs (CHIC-W). One molecule of myosin (223 kDa) consists of two myosin heavy chains (56 kDa), two myosin regulatory light chains (18.8 KDa), two essential light chains, and other fragments [24]. Myosin regulatory light chain 2 was also more abundant in the loin of ewe compared to wether lambs. It has been reported that myosin-2 is more abundant in slow-twitch oxidative type muscles than the fast-twitch glycolytic type in pork [25]. For most mammals, females have more slow oxidative fibres in muscles than males [26]. For cattle, a greater abundance of myosin-2 was associated with tender beef (mean shear force 27.9 N) rather than tough beef (mean shear force 69.6 N) [23]. Another myofibrillar protein, the alpha skeletal muscle isoform X4 of actin, was 12.88-fold more abundant in loins from ewe than wether lambs. The abundance of alpha actin was reported to be 5.5-fold higher in the high-quality beef than low-quality beef, and positively correlated to meat tenderness and redness [27,28]. However, tenderness and colour did not differ (*p* > 0.05) for meat from ewe (mean shear force 26.4 N) lambs compared to wether lambs in the current study (mean shear force 26.4 N) [4]. The differences in the abundance of these myofibrillar proteins may not be large enough to be reflected in differences in meat shear force or colour between ewe and wether lambs slaughtered at 6–8 months old.

Glutathione S-transferase P (GSTP) is the only enzyme in our study that differed between ewe and wether lambs. This enzyme plays a role in mechanisms of cellular detoxification and cellular resistance to oxidative damage by catalysing the conjugation of glutathione to potentially toxic compounds such as reactive oxygen species [29]. Greater abundance of GSTP was found in double-muscled Belgian Texel lambs than Romanov lambs but was not correlated to any meat quality characteristics [29]. This enzyme could have an impact on colour and lipid stability and therefore meat shelf-life [30]. However, these parameters were not measured at different time points during shelf-life in the samples from the current study. Immunoglobulin lambda-1 light chain was more abundant in loins from ewe lambs than wether lambs and is a small polypeptide subunit of an antibody (immunoglobulin). Likewise, immunoglobulin lambda-1 light chain has not been associated with any meat quality traits and any differences in abundance are likely to reflect individual lamb immunological responses [31].

Collagen alpha-3 (VI) chain is a stromal protein in connective tissue, which is the only protein that had greater abundance in loins from wether lambs than ewe lambs in the current study. Similar results were reported by Monteschio et al. [32] where both soluble collagen and total collagen content were greater in loins from wether lambs than ewe lambs, likely due to the increased collagen synthesis in males at puberty [33]. Hopkins et al. [34] reported that measurement of collagen concentration did not explain the variation in shear force and sensory tenderness observed in the meat from lambs. The level of mature and thermally stable crosslinks between collagen molecules, rather than the amount of collagen alpha-3 (VI) chain, are the key factors in collagen-related toughness [35].

### 4.2. Proteome Abundance Differences Due to Diet (Grass-W vs. Chic-W)

Lamb production systems in New Zealand have traditionally relied on perennial ryegrasses. There has, however, been an increase in the use of alternative forage types such as chicory in order to offer a diet with a higher nutritive value than traditional pastures for faster lamb liveweight gains [36]. The chicory diet decreased the abundance of actin and myosin-2, but did not affect the abundance of sarcoplasmic or other proteins. Chicory diets are associated with faster growth rates and lambs that are heavier at slaughter [4,36]. This means that lambs grazed on chicory tend to be more physiologically mature at slaughter. As the degree of maturity increases in production animals, there is an increased proportion of fast glycolytic relative to slow oxidative fibres in the muscle [37]. The decreased abundance of actin, myosin-2, and its myosin regulatory light chain 2 in chicory fed lambs is likely a consequence of a shift towards more fast glycolytic fibres in the loin muscle, which is associated with less amount of these myofibrillar proteins [8,25].

### 4.3. Proteome Abundance Differences Due to Age (Wean-W vs. Chic-W)

Loins from lambs slaughtered at a younger age (4 months old) had a greater abundance of myosin-2, myosin-8, actin and troponin C compared to lambs slaughtered at 6 to 8 months old, which might also be due to the differences in maturity and muscle fibre types. For sheep aged from 4 to 22 months old, the proportion of oxidative fibres in the loin decreased with animal age due to the relatively faster development of glycolytic fibres [38]. Therefore, the proportion of slow oxidative fibres is likely to be lower in the muscle of the older 6- to 8-month-old lambs, which resulted in a lower abundance of myofibrillar proteins, compared with lambs slaughtered at 4 months old in the current study. One protein that differed in meat from WEAN-W and CHIC-W lambs, other than some myofibrillar proteins, was heparan sulfate proteoglycan, which is a heavily glycosylated protein that binds to a variety of protein ligands and regulates a wide range of biological activities, including developmental processes, angiogenesis, blood coagulation, and tumour metastasis [39,40]. The heparan sulfate proteoglycan 2 gene in chromosomes SSC6 that encodes a large proteoglycan was positively associated with marbling score in pork [41]. In the current study, meat from WEAN-W lambs showed a greater abundance of heparan sulfate proteoglycan 2 protein and a higher intramuscular fat percentage than meat from CHIC-W (3.2% vs. 2.6%, respectively) [4]. Further investigation is needed to evaluate whether heparan sulfate proteoglycan could be considered as a potential indicator of fat deposition in meat producing animals.

## 5. Conclusions

In total, 286 proteins were identified across lamb loin samples from six New Zealand forage production systems using liquid chromatography-tandem mass spectrometry. Based on a negative binomial model, most proteins expressed similar abundances, while 16 proteins showed different abundances in meat from the different animal groups. Differences in the abundance of some proteins did not seem to be large enough to impact meat quality (e.g., greater abundance of heat shock protein in muscle from REDC-W lambs or greater myofibrillar proteins in meat from ewes did not translate into lower or higher meat tenderness, respectively). The abundance of creatine kinase was significantly different in meat from MXME-W lambs which tend to have higher ultimate pH than meat from other breeds.

Differences in abundances of myofibrillar proteins were evident due to animal gender, diet and age, sarcoplasmic and stromal proteins due to animal gender, and proteoglycan protein due to animal age. The greater abundance of myofibrillar and lower abundance of stromal proteins in ewes than wethers, were likely due to differences in the proportions of muscle fibre types and collagen. A chicory diet resulted in decreased abundance of some myofibrillar proteins in lamb loins compared to traditional perennial ryegrass, possibly due to greater proportion of muscle glycolytic fibres from faster growing lambs and heavier carcasses at slaughter. Similarly, the abundance of myofibrillar proteins in loins decreased when lambs were slaughtered at 6 to 8 months old compared to 4 months old, which was probably due to faster growth rate of glycolytic fibres than oxidative fibres during the finishing period. Overall, these results provide information on the effect of production factors on the relative abundance of meat proteins contributing to the understanding of the intrinsic determinants that can influence meat quality. Quantification of the contribution of individual proteins in the current study to meat quality characteristics remains a challenge and further investigation is needed to shed light on the role of these proteins.

## Figures and Tables

**Figure 1 foods-11-01419-f001:**
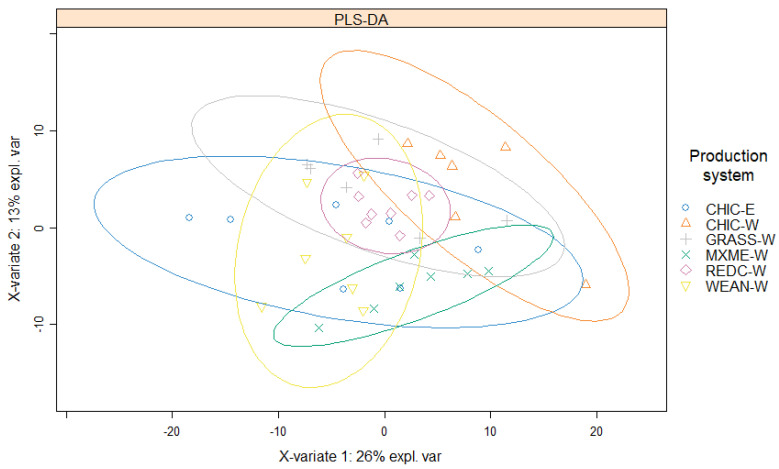
Partial least squares discriminant analysis of all identified proteins in raw lamb loins from six New Zealand commercial production systems (4-month-old wether lambs of a composite breed at weaning, WEAN-W; 6- to 8-month-old wether lambs of a composite breed that had been grazing red clover, REDC-W; 6- to 8-month-old wether lambs of a composite breed that had been grazing perennial ryegrass-based pasture, GRASS-W; 6- to 8-month-old wether and ewe lambs of a composite breed that had been grazing chicory, CHIC-W and CHIC-E; and 12-month-old wether Merino lambs that had been grazing a mixed pasture, MXME-W). *n* = 8 for REDC-W, *n* = 7 for MXME-W, *n* = 6 for GRASS-W, *n* = 6 for CHIC-W, *n* = 7 for CHIC-E, and *n* = 7 for WEAN-W.

**Table 1 foods-11-01419-t001:** Animal sex, diet, genetics, and age at slaughter for Farm A, B, and C (*n* = 8 lambs per group).

Production System	Farm	Sex	Diet ^1^	Genetics	Approximate Age at Slaughter (Months)
REDC-W	A	Wethers	Red Clover	Perendale × LambSupreme ^2^	6–8
MXME-W	B	Wethers	Pasture	Merino	12
GRASS-W	C	Wethers	Pasture	Composite ^3^	6–8
CHIC-W	C	Wethers	Chicory	Composite ^3^	6–8
CHIC-E	C	Ewes	Chicory	Composite ^3^	6–8
WEAN-W	C	Wethers	Pre-weaning	Composite ^3^	4

^1^ Farm A: red clover (*Trifolium pratense*). Farm B: perennial ryegrass (*Lolium perenne*) and white clover (*Trifolium repens*) mix, followed by fescue (*Lolium arundinaceum*), red and white clover, and plantain (*Plantago lanceolata*) mix during the last 2 weeks. Farm C: predominantly Italian (*Lolium multiflorum*) and perennial ryegrass, and red and white clover mix. Farm C-Pre-weaning: suckled and grazing mothers’ diet of a chicory (*Cichorium intybus*) and red clover mix. ^2^ LambSupreme: lean-selected Poll Dorset, Wiltshire, Romney × Dorset, Coopworth, Texel, and high-growth Romney. ^3^ Composite: Perendale, Texel, Finnish Landrace, and Romney.

**Table 2 foods-11-01419-t002:** Identified myofibrillar, sarcoplasmic, stromal, and proteoglycan proteins from raw lamb longissimus lumborum muscle that significantly differed between commercial forage production systems using a negative binomial model. A complete list of all identified proteins is presented as Appendix A. *n* = 8 for REDC-W, *n* = 7 for MXME-W, *n* = 6 for GRASS-W, *n* = 6 for CHIC-W, *n* = 7 for CHIC-E, and *n* = 7 for WEAN-W.

Protein	REDC-W	MXME-W	GRASS-W	CHIC-W	CHIC-E	WEAN-W	Accession Number
Myofibrillar proteins ^1^							
Troponin C, skeletal muscle isoform X1	AB	AB	AB	B	A	A	XP_027832152.1
Actin, alpha skeletal muscle isoform X4	A	A	A	B	A	A	XP_004021390.1
Myosin-2	A	A	A	B	A	A	XP_027830685.1
Myosin-8	A	AB	AB	B	AB	AB	XP_027830687.1
Myosin light chain 1	AB	A	AB	A	B	AB	A0A0H3V384
Myosin regulatory light chain 2, skeletal muscle isoform	A	A	A	B	A	A	NP_001138655.1
Myosin regulatory light chain 2, skeletal muscle isoform isoform X1	A	A	A	B	A	A	XP_011959304.1
Sarcoplasmic proteins ^1^							
Hemoglobin subunit beta	AB	AB	A	AB	B	A	NP_001091117.1
Heat shock protein family A (Hsp70)	A	B	B	B	B	B	W5NPN4
Heat shock cognate 71 kDa protein	A	B	B	B	B	B	XP_011951023.2
Creatine kinase U-type, mitochondrial	B	A	B	B	B	B	XP_011954300.1
Creatine kinase B-type	A	A	AB	B	AB	AB	XP_027813246.1
Immunoglobulin lambda-1 light chain-like	BCD	A	ABC	CD	AB	D	XP_027812629.1
Stromal proteins ^1^							
Collagen alpha-3(VI) chain	AB	AB	AB	A	B	AB	XP_027823071.1
Collagen type VI alpha 3 chain	AB	AB	AB	A	B	AB	W5QCP9
Proteoglycan ^1^							
Proteoglycan protein ^1^							
Heparan sulfate proteoglycan 2	AB	A	ABC	C	BC	AB	W5PEL7

^1^ A, B, C, D Letters within the same row represent the difference in predicted means of the protein between groups. The same letter indicates that there is no significant difference between the groups, while a different letter indicates a significant difference with a *p*-value < 0.05 (Benjamini–Hochberg adjusted). Samples in multiple groups are indicated by more than one letter.

**Table 3 foods-11-01419-t003:** Identified myofibrillar, sarcoplasmic, and stromal proteins that significantly differed between raw lamb longissimus lumborum muscle from 6- to 8-month-old ewe and wether lambs grazed on chicory. *n* = 6 for CHIC-W and *n* = 7 for CHIC-E.

Protein	Fold Change (Ewe/Wether)	Adjusted *p*-Values ^1^	Coverage (%)	Unique Peptides	Accession
Myofibrillar proteins					
Actin, alpha skeletal muscle isoform X4	12.88	<0.001	76	3	XP_004021390.1
Myosin-2	5.48	<0.001	61	3	XP_027830685.1
Myosin regulatory light chain 2, skeletal muscle isoform	5.12	<0.001	85	6	NP_001138655.1
Myosin regulatory light chain 2, skeletal muscle isoform isoform X1	9.27	<0.001	99	3	XP_011959304.1
Troponin C, skeletal muscle isoform X1	2.58	0.038	80	9	XP_027832152.1
Myosin light chain 1	2.61	0.005	78	2	A0A0H3V384
Sarcoplasmic proteins					
Immunoglobulin lambda-1 light chain-like	2.51	0.007	35	2	XP_027812629.1
Glutathione S-transferase P	3.07	0.003	1	2	W5QCP9
Stromal proteins					
Collagen alpha-3(VI) chain	0.33	<0.001	1	2	XP_027823071.1

^1^ *p*-value cut-off of 0.05 with Benjamini-Hochberg correction.

**Table 4 foods-11-01419-t004:** Identified myofibrillar proteins that significantly differed between raw lamb longissimus lumborum muscle from 6- to 8-month-old ewe lambs grazed on perennial ryegrass and chicory. *n* = 6 for GRASS-W and *n* = 6 for CHIC-W.

Protein	Folder Change (Grass/Chicory)	Adjusted *p*-Values ^1^	Coverage (%)	Unique Peptides	Accession
Myofibrillar proteins					
Actin, alpha skeletal muscle isoform X4	17.22	<0.001	76	3	XP_004021390.1
Myosin regulatory light chain 2, skeletal muscle isoform	5.22	<0.001	85	6	NP_001138655.1
Myosin regulatory light chain 2, skeletal muscle isoform isoform X1	6.21	<0.001	99	3	XP_011959304.1
Myosin-2	3.70	<0.001	61	3	XP_027830685.1

^1^ *p*-value cut-off of 0.05 with Benjamini-Hochberg correction.

**Table 5 foods-11-01419-t005:** Identified myofibrillar and proteoglycan proteins that significantly differed (*p* < 0.05) between raw lamb longissimus lumborum muscle from 4-month-old pre-weaning lambs and 6- to 8-month-old wether lambs grazed on chicory. *n* = 7 for WEAN-W and *n* = 6 for CHIC-W.

Protein	Folder Change (Wean/Chicory)	Adjusted *p*-Values ^1^	Coverage (%)	Unique Peptides	Accession
Myofibrillar proteins					
Actin alpha skeletal muscle isoform X4	16.66	<0.001	76	3	XP_004021390.1
Myosin regulatory light chain 2 skeletal muscle isoform	6.25	<0.001	85	6	NP_001138655.1
Myosin regulatory light chain 2 skeletal muscle isoform X1	5.26	<0.001	99	3	XP_011959304.1
Myosin-2	5.26	<0.001	61	3	XP_027830685.1
Myosin-8	2.33	0.032	42	7	XP_027830687.1
Troponin C skeletal muscle isoform X1	2.86	0.002	80	9	XP_027832152.1
Proteoglycan					
Proteoglycan protein					
Heparan sulfate proteoglycan 2	2.77	0.003	1	2	W5PEL7

^1^ *p*-value cut-off of 0.05 with Benjamini–Hochberg correction.

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
