# Peer review of "Proteomic Profile of M. longissimus thoracis from Commercial Lambs Reared in Different Forage Systems"

_foods, 2022, doi:10.3390/foods11101419_

Round 1
Reviewer 1 Report
After reading the manuscript "Proteomic profile of M. longissimus thoracis from commercial lambs reared in different forage systems" a major revision is needed to improve the overall quality of the actual version of the MS. I encourage the authors to revise the full manuscript before the new submission. There is a lack of information about the justificative of the MS (what is the news of this MS? is it going to improve the actual meat industry/chain process?), references, and discussion (the results were poorly discussed in the entire MS). The English need to be revised, I recommend sending the manuscript to a language service.
How did you determine the sample size for this study which is very less?
The author stored meat at -1.5 0C for 21 days and they carried out research for further analysis at -18 0C what is this.
Author Response
After reading the manuscript "Proteomic profile of M. longissimus thoracis from commercial lambs reared in different forage systems" a major revision is needed to improve the overall quality of the actual version of the MS.
I encourage the authors to revise the full manuscript before the new submission. There is a lack of information about the justificative of the MS (what is the news of this MS? is it going to improve the actual meat industry/chain process?)
Response: The label free proteomics mass spectrum quantification used in this study is not unique, however, to the best of our knowledge, there are very few studies that investigated the proteomic profile of lamb/sheep meat. With a greater understanding of how commercial on-farm factors affect the proteomics/meat quality profile of lamb meat, farmers and the meat industry can potentially improve the quality of lamb. The justification of the study is included and discussed in the Introduction section and has been re-emphasized.
references, and discussion (the results were poorly discussed in the entire MS). The English need to be revised, I recommend sending the manuscript to a language service.
Response: Several authors are native English speakers and therefore a language service is not required. However, we have clarified the wording and elaborated further on the results.
How did you determine the sample size for this study which is very less?
Response: The 6 lamb groups were selected to represent the most common New Zealand commercial production systems. The number of lambs in each group was determined to give enough representation of lambs from those production systems. A greater sample size can always provide greater representativeness of the variation in protein profile, however, the sample size utilised was determined from balancing the considerations of power analysis versus proteomics studies being expensive and time consuming.
The author stored meat at -1.5 0C for 21 days and they carried out research for further analysis at -18 0C what is this.
Response: It is a commonly implemented storage method for vacuum-packed lamb from New Zealand as it allows for proteolysis which is known as aging. Aging is a natural process that improves tenderness and flavour of meat under refrigerated conditions. This is a very common practice in the meat industry in New Zealand and internationally and it does not need further explanation, however, we have made it clear by adding the words “for aging”.
Reviewer 2 Report
L 83- why did you aged the meat for 21 days?
L 83 - please explain in short your sample collection
Table 1 - please explain why you did not have the appropriate combinations of age, sex, and age in the study? eg you compared sex...but you had comparison only among CHIC-W and CHIC-E group
L 86 -please explain the parameters involved during freeze-drying process
L 89 - freeze dried samples were grounded and again frozen? Under which conditions?
L 298 -could you explain the possible reason of this result?
l 346 -what do you think about the fact that you previously aged the meat samples for 21 days?
Conclusion - please conclude, do not repeat and summary again the results
Author Response
L 83- why did you aged the meat for 21 days?
Response: Aging of carcasses and cuts is a natural process that improves tenderness and flavour of meat under refrigerated conditions. This is a common practice by the meat industry in New Zealand and we wanted to follow commercial process for this meat.
L 83 - please explain in short your sample collection
Response: Further information about sample collection has been added.
Table 1 - please explain why you did not have the appropriate combinations of age, sex, and age in the study? eg you compared sex...but you had comparison only among CHIC-W and CHIC-E group
Response: This paper is part of a larger study that aimed to investigate meat quality differences between major New Zealand commercial lambs. The key aspect is that it represents commercial types of animals produced in New Zealand for lamb. Therefore, the treatments represent what is coming off farm, and it is not possible to get all the cross comparisons. Eg only the Merino lambs are normally slaughter at older (12-month) age, and it’s not available to have a 6–8-month-old comparison group from commercial farms. We have tried to make this more clear in the manuscript.
L 86 -please explain the parameters involved during freeze-drying process
Response: Meat was freeze dried at −50 °C for 2 days. This has been added to the manuscript.
L 89 - freeze dried samples were grounded and again frozen? Under which conditions?
Response: The samples were freeze dried first, and then grounded into fine powder in room temperature. The grounded samples were stored in small vials fulfilled with nitrogen and stored in −80 °C freezer before protein extraction. We have added further details to the manuscript.
L 298 -could you explain the possible reason of this result?
Response: It is unknown as to why the red clover treatment resulted in greater heat shock proteins. This is something that would require further investigation and so we have indicated this in the manuscript. Heat shock proteins can function as molecular chaperones, facilitating protein folding, preventing protein aggregation, or targeting improperly folded proteins to specific degradative pathways.
l 346 -what do you think about the fact that you previously aged the meat samples for 21 days?
Response: Aging may influence the relative concentration of some proteins, however, all samples in the current study were aged using the same conditions, and the focus of this study was the proteomics profile and meat quality of aged meat product which is representative of the meat industry in New Zealand.
Conclusion - please conclude, do not repeat and summary again the results
Response: Conclusions were revised.
Reviewer 3 Report
The work is eminently scientific and written at a very high level. It is written clearly, and the results were discussed and compared with those obtained by other authors.
The Authors took the influence of several factors affecting the meat's quality into account in the context of the different protein fractions found in the Ld muscle.
The question arises whether it is possible to present the conclusions of the work also in a model. Specialists and meat technologists could easier understand thanks to it.
A potential model could be easier to notice what kinds of proteins are more and minor and what it means.
In addition, such a model could be an inspiration in the following stages of research that the Authors intend to continue.
Author Response
The work is eminently scientific and written at a very high level. It is written clearly, and the results were discussed and compared with those obtained by other authors.
The Authors took the influence of several factors affecting the meat's quality into account in the context of the different protein fractions found in the Ld muscle.
The question arises whether it is possible to present the conclusions of the work also in a model. Specialists and meat technologists could easier understand thanks to it.
A potential model could be easier to notice what kinds of proteins are more and minor and what it means.
In addition, such a model could be an inspiration in the following stages of research that the Authors intend to continue.
Response: Thank you for this suggestion which is a great idea to incorporate in the future as we collect more data. A modelling approach would not be suitable with this limited data set (one time point collection), beyond the statistical models that we already applied. This data set could be used as part of a meta-analysis that considers a range of studies providing an opportunity to build better links between the level of proteins and meat quality characteristics. We would keep this in mind for our future work, since such models can be built with further data collection.
Reviewer 4 Report
The aim and design of the study are given clearly. The study is of a quality that will contribute to science.
Author Response
The aim and design of the study are given clearly. The study is of a quality that will contribute to science.
Response: Thank you for your comments
Reviewer 5 Report
Foods-1687966
Proteomic profile of M. longissimus thoracis from commercial 2 lambs reared in different forage systems
The authors of the manuscript did an excellent job with the design, identifying the breed, age, and food regimen on the effect on the quality of lamb meat. The manuscript is very well structured; the clarity of the results defines the success of the experimental proposal. Likewise, the manuscript had been rigorously and carefully written; facilitating the message and capturing the reader. Congratulations.
There are some small mistakes, and suggestions for improvements to the manuscript, which can be identified as highlighted text in the pdf file. The suggestions are presented below:
Line 83
It said: -1.5°C for 21 days then frozen at -20°C until
It should say: -1.5 °C for 21 days then frozen at -20 °C until
Line 89
It said: (7 M Urea,
It should say: (7M Urea,
Line 90
It said: Thiourea,1% freshly-
It should say: Thiourea, 1% freshly-
Line 109
It said: in 0.1% formic acid prior
It should say: in 0.1% formic acid (FA) prior
Line 118
It said: 0.1% Formic acid;
It should say: 0.1% FA;
Line 331
It said: (223kDa)
It should say: (223 kDa)
Line 332
It said: (56kDa)…… (18.8KDa),
It should say: (56 kDa)…… (18.8 KDa),
Line 347
It said: Glutathione S-transferase P
It should say: Glutathione S-transferase P (GSTP)
Line 351
It said: Glutathione S-transferase P
It should say: GSTP

Author Response
The authors of the manuscript did an excellent job with the design, identifying the breed, age, and food regimen on the effect on the quality of lamb meat. The manuscript is very well structured; the clarity of the results defines the success of the experimental proposal. Likewise, the manuscript had been rigorously and carefully written; facilitating the message and capturing the reader. Congratulations.
Response: Thank you for the positive feedback
Round 2
Reviewer 1 Report
The overall quality of the article was improved than the previous one therefore it may be accepted.
Reviewer 2 Report
The research is not conducted correctly and author replies to my comments are not enough clear.